# Intercellular Transmission of Naked Viruses through Extracellular Vesicles: Focus on Polyomaviruses

**DOI:** 10.3390/v12101086

**Published:** 2020-09-26

**Authors:** Francois Helle, Lynda Handala, Marine Bentz, Gilles Duverlie, Etienne Brochot

**Affiliations:** 1UR UPJV 4294, Agents Infectieux, Résistance et chimiothérapie (AGIR), Centre Universitaire de Recherche en Santé, Université de Picardie Jules Verne, 80000 Amiens, France; l.handala@gmail.com (L.H.); marinebentz@yahoo.fr (M.B.); gilles.duverlie@u-picardie.fr (G.D.); etienne.brochot@u-picardie.fr (E.B.); 2Laboratoire de Virologie, Centre Hospitalier Universitaire, 80000 Amiens, France

**Keywords:** Polyomavirus, BKPyV, JCPyV, MCPyV, TSPyV, SV40, extracellular vesicles, en bloc transmission, neutralizing antibodies

## Abstract

Extracellular vesicles have recently emerged as a novel mode of viral transmission exploited by naked viruses to exit host cells through a nonlytic pathway. Extracellular vesicles can allow multiple viral particles to collectively traffic in and out of cells, thus enhancing the viral fitness and diversifying the transmission routes while evading the immune system. This has been shown for several RNA viruses that belong to the Picornaviridae, Hepeviridae, Reoviridae, and Caliciviridae families; however, recent studies also demonstrated that the BK and JC viruses, two DNA viruses that belong to the Polyomaviridae family, use a similar strategy. In this review, we provide an update on recent advances in understanding the mechanisms used by naked viruses to hijack extracellular vesicles, and we discuss the implications for the biology of polyomaviruses.

## 1. Introduction

Extracellular vesicles (EVs) are lipid bilayer-delimited particles that are physiologically released from cells and carry proteins, nucleic acids, lipids, or metabolites to recipient cells [1]. They are produced via different mechanisms (Figure 1): (i) direct budding from the plasma membrane to form microvesicles, also called ectosomes; (ii) budding of intraluminal vesicles (ILVs) during the process of multivesicular body (MVB) formation and release of exosomes in the extracellular environment as a result of the fusion of MVBs with the plasma membrane; (iii) autophagosome-mediated exit without lysis or secretory autophagy that releases single-membrane vesicles after the fusion of double-membraned autophagosomes with the plasma membrane; (iv) apoptosis that generates apoptotic bodies. Exosomes are generally small in size (50–150 nm), while microvesicles and secretory autophagy-derived EVs range from 100 to 1000 nm. Apoptotic bodies are generally larger than 1 µm.

Naked viruses have long been considered to be released from infected cells after cell lysis. However, in the last decade, several studies evidenced that various naked RNA viruses hijack EVs to exit host cells through a nonlytic pathway. This includes several members of the Picornaviridae family and the *Enterovirus* genus (Poliovirus [2,3], Coxsackievirus B3 [3,4], Rhinovirus [3], Enterovirus 71 [5,6]), the *Cardiovirus* genus (Encephalomyocarditis Virus [7]), and the *Hepatovirus* genus (hepatitis A virus (HAV) [8]), as well as viruses that belong to other families such as the hepatitis E virus (HEV; Hepeviridae) [9], Rotavirus (Reoviridae) [10], or Norovirus (Caliciviridae) [10]. EV hijacking may confer several advantages to naked viruses: (i) the possibility to be released from infected cells through a nonlytic pathway; (ii) a diversification of transmission routes which promotes the propagation; (iii) an enhancement of the viral fitness thanks to en bloc delivery and genetic cooperativity; (iv) a protection against neutralizing antibodies that target the viral capsid [11,12,13,14].

Polyomaviruses are small nonenveloped DNA viruses that can cause various diseases in birds and mammals, including humans [15]. Human polyomaviruses are quite harmless in immunocompetent individuals, but they are associated with several diseases in immunocompromised patients. The JC Polyomavirus (JCPyV) can cause progressive multifocal leukoencephalopathy (PML) in acquired immune deficiency syndrome (AIDS) patients, as well as in individuals under immunomodulatory biotherapy [16]. The BK Polyomavirus (BKPyV) is associated with nephropathy in renal transplant recipients and late-onset hemorrhagic cystitis in recipients of hematopoietic stem-cell transplantation [17]. The Merkel Cell Polyomavirus (MCPyV) can cause Merkel cell carcinoma, an aggressive type of skin cancer [18], and the Trichodysplasia Spinulosa Polyomavirus (TSPyV) is associated with trichodysplasia spinulosa, a rare cutaneous condition [19]. Since 1989, it has been suggested that polyomaviruses could be released from infected cells without cell lysis [20,21]. Furthermore, very recently, we and the Atwood Laboratory demonstrated that the BKPyV and JCPyV use EVs to infect target cells [22,23,24]. Whether other polyomaviruses are also released within EVs is not yet known, but this is very likely. Becker et al. also observed MCPyV particles tightly associated with a double-layer lipid membrane in endosomal compartments [25]. However, the authors concluded that this membrane was acquired during endosomal trafficking since these enveloped particles were neither observed in the viral inoculum nor during the early entry steps. This review is aimed at providing an update on recent advances in understanding the mechanisms used by naked viruses to hijack EVs, and we discuss implications for the biology of polyomaviruses.

## 2. Release of EV-Associated Virions

As mentioned above, several mechanisms enable the release of extracellular vesicles, and it has been shown that the different EV production pathways can be exploited by naked viruses for their release (Figure 1). The release of HAV, HEV, and Norovirus virions has been shown to be related to the exosomal pathway [8,9,10]. Indeed, membrane-associated HEV particles were observed within MVBs by electron microscopy [9]. Furthermore, HEV release was increased by treatment with bafilomycin A1, an accelerator of exosome release due to lysosomal inhibition, and it was decreased by treatment with GW4869, which inhibits ceramide biosynthesis and, thus, exosome release, or by depletion of Rab27A [9]. Moreover, the protein content of EV-associated HAV is highly enriched in endolysosomal components and lacks markers of autophagy, confirming that an exosome-like mechanism egress is involved in endosomal budding of HAV capsids into MVBs [26]. In contrast, it has been suggested that Rotavirus particles egress from cells nonlytically in large microvesicles derived from the plasma membrane [10] and an autophagosome-mediated release has been evidenced for EV-associated Poliovirus and other enteroviruses. Stimulation of autophagic processes increases Poliovirus spread, while inhibition of autophagy reduces its spread [2,27,28]. Furthermore, Poliovirus and other enteroviruses are released in vesicles that are enriched in phosphatidylserine and contain lipidated LC3-II, a marker of autophagy [3,4,28].

The mechanism involved in the release of EV-associated JCPyV and BKPyV has not yet been investigated. For EV-associated JCPyV, Santiana et al. commented that the presence of CD9, CD81, flotillin-1, annexin-V, and TSG-101 proteins in the EVs suggests either an MVB or plasma-membrane origin [29]. However, these markers might be associated with unrelated EVs with similar buoyant density. Since small EVs containing a few particles were observed for JCPyV [23,24], whereas we observed large EVs carrying one or more tens of viral particles for BKPyV [22], it is likely that both viruses use different mechanisms for their nonlytic release. Another possibility is that they may be released by infected cells in multiple EV subtypes at distinct time points during infection, as recently described for the Poliovirus and Encephalomyocarditis Virus [7,30].

The endosomal sorting complex required for transport (ESCRT) machinery buds membranes and severs membrane necks from their inner face and, thus, plays a major role in the biogenesis of EVs [31,32]. It is also recruited for the budding of enveloped viruses by means of Pro-rich “late domain” motifs in structural proteins: PPXY, P(S/T)AP, or (L)YPX_1/3_L [33]. Several lines of evidence indicated that the biogenesis of EV-associated HAV and HEV also depends on components of the ESCRT machinery. HAV structural proteins VP2 and pX interact with the ESCRT-associated protein ALIX, and knockdown of the ALIX protein expression inhibits the release of EV-associated HAV from infected cells [8,34,35]. On the other hand, EV-associated HEV egress relies on the ESCRT machinery through the interaction between the ORF3 viral protein and the TSG-101 ESCRT protein [36,37]. The role of the ESCRT machinery in the polyomavirus life cycle has not yet been investigated, but it is interesting to note the presence of a potential late domain conserved in the VP1 sequence of several polyomaviruses [38,39].

In addition to recruitment of the ESCRT machinery, other function of viral proteins can be required for virion release. For instance, Ding et al. demonstrated that ORF3′s ion-channel activity is important for the release of HEV infectious particles [40]. Interestingly, immediately upstream of the VP1 gene of SV40, BKPyV, and JCPyV polyomaviruses resides a gene referred to as agnogene that encodes for an approximately 70 amino acid (aa) long Agnoprotein [41]. Even though the precise role of the Agnoprotein remains unclear, it seems to be involved in virus egress [42], and a viroporin activity was described [43]. We hypothesized that the Agnoprotein could play a role in EV-associated polyomavirus egress. However, this protein does not seem to be essential for virion envelopment since EV-associated JCPyV pseudovirions can be released in the absence of the Agnoprotein [23].

## 3. Entry of EV-Associated Virions

Several reports indicated that naked and EV-associated virions can use alternative pathways to enter target cells. Indeed, molecules involved in the uptake of EVs [44] can serve that of EV-associated virions. In particular, the phosphatidylserine lipids (PS) which are found on the EV outer membrane leaflet can facilitate virion endocytosis by associating with PS receptors on recipient cells. Thus, contrary to what was thought previously, TIM1 is not an essential HAV entry factor but its PS-binding activity contributes to the spread of enveloped virions [45]. PS also seems to be involved in the uptake of EV-associated enteroviruses, Rotavirus, and Norovirus [3,10]. However, even though distinct intracellular trafficking routes are used by naked and EV-associated virions, infection can still require binding to the specific virus receptor. This has been demonstrated for the HAV, Poliovirus, Coxsackievirus, Rhinovirus, or murine Norovirus [3,10,46], which suggests that EV membranes get disrupted following endocytosis, possibly by endosomal lipases and lipid extractor proteins. Consistent with this hypothesis, depletion and pharmacological inhibition of the cholesterol transporter Niemann–Pick disease type C1 (NPC1) protein and the lysosomal acid lipase (LAL), significantly impaired EV-associated HAV and HEV but not naked HAV and HEV infection [47,48].

Naked JCPyV virions enter cells via a two-step mechanism in which the VP1 viral capsid protein specifically attaches to host cells via the sialic acid moiety of lactoseries tetrasaccharide C (LSTc) [16,49] and then binds one member of the 5-hydroxytryptamine (5HT2) family of serotonin receptors for internalization [16,50,51]. An alternative entry pathway involving a heparin-like attachment receptor and a nonsialylated coreceptor has also been described [52]. Concerning BKPyV, naked particles interact with the sialic-acid moiety of different types of b-series gangliosides such as GD3, GD2, GD1b, and GT1b, but an N-linked glycoprotein containing (2,3)-linked sialic acid could also act as a host receptor [53,54,55]. Both polyomaviruses are then internalized by endocytosis, followed by viral trafficking to the endoplasmic reticulum (ER) and nucleus for replication. We and the Atwood laboratory clearly demonstrated that EV-associated BKPyV and JCPyV use alternative pathways to enter target cells, as compared to naked particles [22,23,24]. Indeed, neuraminidase treatment, which removes cell-surface sialic acids, efficiently inhibits naked virion entry, whereas it only has a slight effect on infection with enveloped virions [22,23,24]. It was also shown that infection by EV-associated JCPyV does not depend on the presence of 5HT2 receptors since membrane-wrapped particles are able to infect 5HT2-null cells [23]. Furthermore, JCPyV pseudovirions harboring the VP1 sialic-acid- and LSTc-binding-pocket mutations L54F and S268F can transduce cells when associated with EVs but not when purified as naked particles [23]. O’Hara et al. also demonstrated that EV-associated JCPyV uptake was dependent on both macropinocytosis and clathrin-dependent endocytosis [24]. Using electron microscopy, we observed the presence of intact vesicles carrying BKPyV virions in endosomal compartments, confirming that virions can hijack molecules involved in the uptake of EVs to enter target cells [22]. As other EV-transported naked viruses, it seems likely that EV membranes are disrupted by endosomal lipases and lipid extractor proteins to release free virions in the endosomes enabling EV-associated virions and naked virions to use a similar mechanism to traffic from endosomes to the ER.

## 4. Enhancement of Viral Fitness/Genetic Cooperativity

Electron microscopy indicated that enveloped HEV particles are around 40 nm in diameter and that the capsids of HEV particles are individually covered by lipid membranes, similar to enveloped viruses [56]. However, for other viruses, EVs can carry several virions: one to four in the case of HAV [8], one or more tens in the case of enteroviruses [3] or Rotavirus [10]. This allows multiple viral genomes to be collectively transferred into target cells, thus locally increasing the multiplicity of infection and rendering infection more efficient as compared to individual viral particles [13,14]. This has been demonstrated in vitro for the Poliovirus, Rotavirus, and Norovirus [3,10], as well as in vivo for the Rotavirus [10]. The Altan-Bonnet group also proposed that this could facilitate genetic cooperativity among deleterious mutants [13,14]. However, it was recently observed, at least for Coxsackievirus B3, that EV-associated virions undergo intercellular transmission as pools of sibling viral genomes [57]. Hence, if cooperation occurs, it should probably involve viral particles derived from the same parental genome rather than different variants.

As mentioned previously, EVs containing a few particles were observed for JCPyV [23,24], whereas we observed EVs carrying one or more tens of viral particles for BKPyV [22]. O’Hara et al. indicated that EVs derived from infected choroid plexus epithelial cells were much more infectious than purified JCPyV particles [24]. In contrast, we did not observe such an enhancement in viral fitness for BKPyV since the infectivity of EV-associated virions was similar before and after treatment of EVs with chloroform [22]. However, we used the Dunlop strain, which contains a rearranged noncoding control region and is highly adapted to cell culture, and it would be interesting to investigate a potential enhancement in the viral fitness in the context of a BKPyV archetypal strain.

## 5. Neutralization of EV-Associated Virions

Membrane hijacking also represents an elegant strategy for evading neutralizing antibodies that target the viral capsids. In particular, this was demonstrated for HAV [8] and HEV [56,58]. However, it was also shown that HAV particles within vesicles become sensitive to neutralizing antibodies after engagement with the host cell (as late as 6 h after inoculation), supporting the model of vesicle membrane disruption within an endocytic compartment [8]. This implies that neutralizing antibodies are also internalized, either nonspecifically or through a specific mechanism (e.g., Fc-mediated), to interact with viral capsids once the vesicle membrane has been disrupted.

Interestingly, whereas EV-associated JCPyVs are resistant to JCPyV-specific antisera [23,24], we did not observe any difference in the sensitivity of enveloped BKPyV and naked virions to BKPyV-specific antisera and commercially available polyvalent immunoglobulin preparations [22]. We also observed, in a time-of addition assay, that enveloped BKPyVs, as well as naked virions, were neutralized when cells were exposed to polyvalent immunoglobulin preparations either immediately before inoculation with virus or up to 4 h afterward. This difference between JCPyV and BKPyV could suggest that different entry steps are targeted by neutralizing antibodies for each virus.

## 6. Consequences for the Physiopathology of Polyomaviruses

Naked and EV-associated virions may play distinct but equally important roles in viral pathogenesis. For instance, naked virions may be responsible for indirect transmission of viruses between individuals, whereas EV-associated virions may facilitate subsequent spread within infected hosts. Furthermore, cell lysis within the body is a highly inflammatory event that triggers immune responses. Thus, it would be an advantage for any virus to be released from infected cells through a nonlytic pathway. PS are also well-known suppressors of inflammation and, thus, EV membranes may play an important role in modulating the immune system [13,14,59].

Similarly, such mechanisms likely contribute to the asymptomatic persistence of BKPyV and JCPyV in immunocompetent individuals. EV hijacking may also be involved in the pathogenesis of these two polyomaviruses. For instance, it could explain how oligodendrocytes and astrocytes, the main targets of JCPyV in the central nervous system (CNS), are infected in PML patients, whereas these cells do not express LSTc [24]. It could also explain why JCPyV strains found in the CNS of PML patients accumulate mutations in the VP1 LSTc-binding pocket, rendering these viruses incapable of binding the LSTc sialic-acid moiety [16,60,61]. Regarding BKPyV, it is absent from the blood of immunocompetent hosts, but low-level virus shedding was demonstrated in the urine of 10% of BKPyV seropositive healthy blood donors, in the absence of any kidney injury and symptom [62]. This could be explained by a polarized secretion of EV-associated BKPyV, as described for HAV [63]. In contrast, BKPyV viremia can be observed in immunocompromised patients, as a consequence of basement membrane rupture induced by virus replication.

## 7. Concluding Remarks and Future Directions

The first two human polyomaviruses, JCPyV and BKPyV, were isolated in 1971 [64,65]. For 50 years, these viruses were considered as naked particles released by cell lysis, but we and the Atwood Laboratory recently described that they also hijack EVs to be released through a nonlytic pathway and to be transmitted en bloc to target cells. On the basis of this new data, we present a model of the BKPyV and JCPyV life cycle in Figure 2. Investigations into the role of EVs in viral transmission are still in their infancy, and many important questions remain unanswered. Concerning BKPyV and JCPyV, it will be important to confirm that EV-associated virions also exist in vivo. It will also be interesting to investigate whether other members of the Polyomaviridae family exploit EVs for their transmission. This could lead to the identification of druggable targets to block polyomavirus replication [66,67].

## Figures and Tables

**Figure 1 viruses-12-01086-f001:**
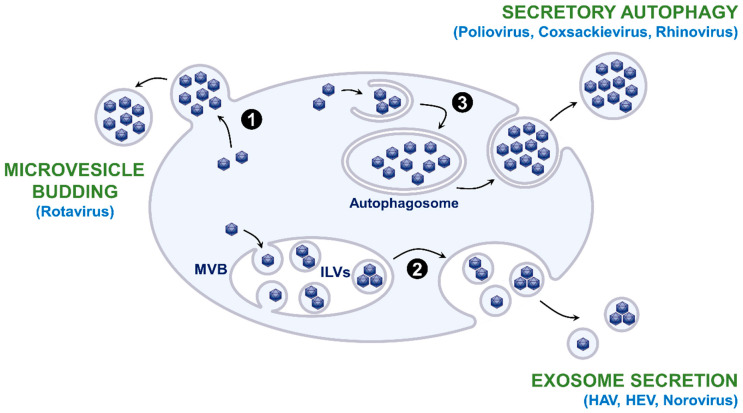
Origin of extracellular vesicles (EVs) and EV hijacking by naked viruses. EVs can directly bud at the plasma membrane as microvesicles (**1**). They can also bud into multivesicular bodies (MVBs) and form intraluminal vesicles (ILVs) that are released as exosomes as a result of the fusion of MVBs with the plasma membrane (**2**). EVs can also be released after the fusion of double-membraned autophagosomes with the plasma membrane, a mechanism termed secretory autophagy (**3**). Apoptosis can also generate EVs called apoptotic bodies, but this mechanism is not shown in the figure. Examples of naked viruses that exploit these different EV production pathways are mentioned in light blue.

**Figure 2 viruses-12-01086-f002:**
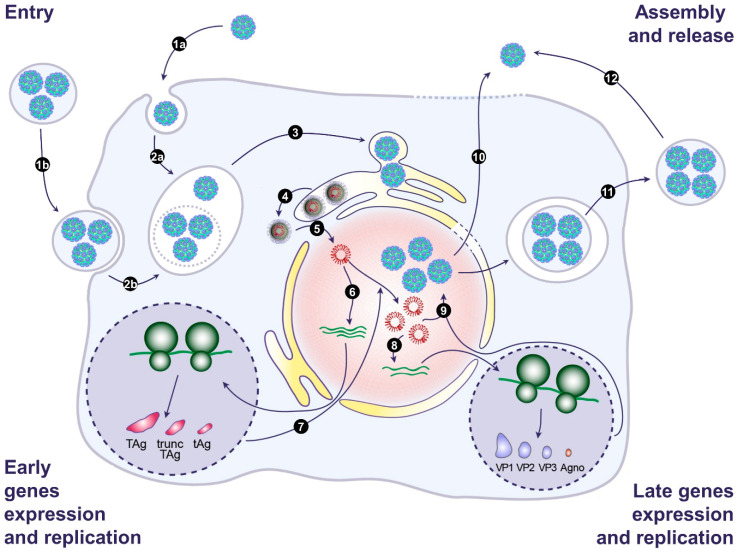
Model of the BK and JC polyomavirus life cycle. (**1**) Infection begins with binding of naked virions (**1a**) or EV-associated virions (**1b**) to the cell surface. This is followed by endocytosis and transport to the endosomes (**2a** and **2b**). EV membranes may be disrupted by endosomal lipases and lipid extractor proteins to release free virions in the endosomes. Then, viral particles traffic from the endosomes to the endoplasmic reticulum (ER) (**3**). In the ER, partial capsid uncoating occurs, which creates a hydrophobic surface exposing VP2/VP3 that integrates into the ER membrane and leads to the release of partially uncoated viruses into the cytosol (**4**). The viral genome is then transported into the nucleus via the nuclear pore complex thanks to VP2/VP3 nuclear localization signals and the importin α/β1 import pathway (**5**). Then, early genes are expressed (**6**), and early proteins are translocated into the nucleus to initiate viral DNA replication (**7**). Late genes are then expressed (**8**), and late proteins are translocated into the nucleus. After translocation, VP1, VP2, and VP3 self-assemble to form capsids into which newly synthetized double-stranded viral DNA is packaged (**9**). Progeny virions are released from infected cells as naked virions after cell lysis (**10**) or through a nonlytic pathway generating EV-associated virions (**11**). Free naked viral particles can also derive from EVs, as a result of spontaneous breakage (**12**). Adapted from [55].

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
