# Peer review of "Intercellular Transmission of Naked Viruses through Extracellular Vesicles: Focus on Polyomaviruses"

_viruses, 2020, doi:10.3390/v12101086_

Round 1

Reviewer 1 Report

Date: September 2, 2020

Title: Intercellular transmission of naked viruses through extracellular vesicles: Focus on polyomaviruses

Authors: Helle et al.,

Journal: Viruses

Reviewer’s Comments

  1. This is a considerably short review paper considering a highly significant topic with respect to the spread of the viral infections via the extracellular vesicles (EVs). The content of this review could be nicely expanded by including various virus groups (DNA and RNA) that employ the EVs as vehicle for their spread. Subsequently, the polyomavirus family would then be discussed under a subtopic due to the fact that a few number of research articles are published regarding EVs and polyomavirus combination.

  1. There are various sections in the paper, which lack referencing (example, lines 56-64)

  1. In addition, “non-published data” was referenced in a couple of lines (example, lines 69, 123-125), which should be avoided to comply with the Journal’s (Viruses) regulations.

  1. Moreover, there are various references that do not properly reflect the original papers but rather refer to the review articles, which could be easily corrected.

Author Response

We thank the reviewer for his comments. We agree that the field of viral spread via EVs is highly significant. We are aware that the review could have been expanded by including various virus groups that employ the EVs as vehicle but as mentioned in the title and the abstract of the review, we voluntarily focused on naked viruses. Besides, it is worth to notice that, to our knowledge and to date, BKPyV and JCPyV are the only two naked DNA viruses known to hijack EVs for their propagation.

As recommended by the reviewer, we added several references in the manuscript and we removed all assumptions that were supported by non-published results.

Reviewer 2 Report

        The review deals with an interesting topic of transmission of whole virus particles in extracellular vesicles. It summarizes new findings concerning the transmission of small RNA viruses, whose morphogenesis takes place in the cytoplasm. Extracellular vesicle-mediated transmission has been observed during infection by  two small DNA viruses assembled in the nucleus, human polyomaviruses JC (JCPyV) and BK (BKPyV) by W. Atwood's group and the reviewers' laboratory, respectively.
Practically nothing is known about the mechanism of transmission of polyomaviruses by extracellular vesicles. Thus, the review speculates about it mostly based on the findings obtained for RNA viruses.

Comments: 

Authors relatively often cite reviews instead of original papers, including the citation of their own previous review (citation 47).

Author Response

We thank the reviewer for his comments. As suggested by the reviewer, we added several references in the manuscript.

Reviewer 3 Report

This is a well-written, concise, and authoritative review on the role of extracellular vesicles in virus dissemination. It is a timely and much needed review particularly for those studying polyomaviruses.

Author Response

We thank the Reviewer for appreciating our work and supporting it for publication.

Round 2

Reviewer 1 Report

Authors fairly addressed the concerns of this reviewer.